# Digital Health in Cardiac Rehabilitation and Secondary Prevention: A Search for the Ideal Tool

**DOI:** 10.3390/s21010012

**Published:** 2020-12-22

**Authors:** Maarten Falter, Martijn Scherrenberg, Paul Dendale

**Affiliations:** 1Heart Centre Hasselt, Jessa Hospital, 3500 Hasselt, Belgium; martijn.scherrenberg@jessazh.be (M.S.); paul.dendale@jessazh.be (P.D.); 2Mobile Health Unit, Faculty of Medicine and Life Sciences, Hasselt University, 3500 Hasselt, Belgium; 3KU Leuven, Faculty of Medicine, 3000 Leuven, Belgium

**Keywords:** digital health, mobile health, sensors, cardiovascular rehabilitation, secondary prevention, heart failure, artificial intelligence, telerehabilitation

## Abstract

Digital health is becoming more integrated in daily medical practice. In cardiology, patient care is already moving from the hospital to the patients’ homes, with large trials showing positive results in the field of telemonitoring via cardiac implantable electronic devices (CIEDs), monitoring of pulmonary artery pressure via implantable devices, telemonitoring via home-based non-invasive sensors, and screening for atrial fibrillation via smartphone and smartwatch technology. Cardiac rehabilitation and secondary prevention are modalities that could greatly benefit from digital health integration, as current compliance and cardiac rehabilitation participation rates are low and optimisation is urgently required. This viewpoint offers a perspective on current use of digital health technologies in cardiac rehabilitation, heart failure and secondary prevention. Important barriers which need to be addressed for implementation in medical practice are discussed. To conclude, a future ideal digital tool and integrated healthcare system are envisioned. To overcome personal, technological, and legal barriers, technological development should happen in dialog with patients and caregivers. Aided by digital technology, a future could be realised in which we are able to offer high-quality, affordable, personalised healthcare in a patient-centred way.

## 1. Introduction

In recent years, digital health has become more incorporated in daily live in general, and in clinical practice in medicine. While electronic health records, smartphone health applications and smartwatches are already becoming part of routine practice, evolutions in the field of telemedicine, robotics and artificial intelligence indicate that our current methods are just the tip of the iceberg [1,2,3]. In cardiology, the playing field is already moving from the hospital to the patients’ homes. Large trials continue to show positive results in the field of telemonitoring via cardiac implantable electronic devices (CIEDs) [4], monitoring of pulmonary artery pressure via implantable devices [5], and telemonitoring via home-based non-invasive sensors [6,7]. Large-scale screening for atrial fibrillation (AF) via already available smartphone technology was shown as feasible in the Apple Heart study [8].

Secondary prevention in cardiac patients, in particular ischemic heart disease and heart failure, is of uttermost importance to reduce morbidity and mortality. Rehabilitation programs are considered a Class Ia indication in the ESC Guidelines [9]. EUROASPIRE V has shown that a majority of coronary patients in Europe has inadequate risk factor control [10] and that inclusion rates in cardiac rehabilitation programs are low [11]. There are well-known barriers in cardiac rehabilitation and secondary prevention on the physician side, such as relatively low guideline adherence [12] and low patient referral to cardiac rehabilitation [11] as well as on the patient side, such as transport distance to a cardiac rehabilitation facility and low medication adherence [13]. Sensors and digital tools, such as decision support tools, motivational and educational smartphone platforms, telemonitoring and telerehabilitation programs can offer solutions for these barriers [14].

The aim of this narrative review is to offer a perspective on the current use of digital technology in cardiac rehabilitation and secondary prevention. As most benefits of cardiac rehabilitation have been shown in patients with ischemic heart disease and heart failure [9,15], the focus of this paper will be on these patient populations. The viewpoint also offers a vision of what ideal secondary prevention and rehabilitation aided by digital tools might look like in the future.

## 2. Methods

This viewpoint is written as a narrative review. The databases PubMed, Embase, and Web of Science were used. Keywords were defined corresponding to the subjects discussed and included cardiac rehabilitation, heart failure, digital health, sensors, among other keywords for the specific subtopics.

## 3. Sensors in Cardiac Rehabilitation and Secondary Prevention

This paper will focus on care for patients with established heart disease. Current care for patients with heart disease consists of risk stratification and risk factor control, patient education, optimalisation of pharmacologic therapy, increasing physical activity, and psychosocial guidance [9]. In psychosocial management, techniques such as breathing exercises and meditation are often included [16]. All of these aspects can be addressed in a cardiac rehabilitation program, which consists of patient assessment, physical activity counselling, exercise training, dietary and nutritional counselling, weight control management, lipid management, blood pressure management, smoking cessation and psychosocial management [17]. While telerehabilitation and telemedicine are not yet considered standard-of-care, current sensors already offer possibilities for application in many domains of secondary prevention.

### 3.1. Hypertension

High blood pressure (BP) is a leading risk factor for heart disease. While in the past BP measurement was mainly performed in the doctor’s office, current practice has largely evolved to home-measurement of BP by patients using digital upper-arm sphygmomanometers. These digital BP monitors measure BP by oscillometric measurements [18].

Aims in development of BP monitors are to increase convenience and accuracy while offering continuous rather than intermittent BP estimation [19]. Already, devices that measure BP at the wrists or fingers through oscillometry are commercially available. However, while progress is being made, accuracy is currently still considered inferior to conventional upper-arm BP monitors and only the use of validated upper-arm BP monitors is recommended in the 2020 American Heart Association Guidelines on Hypertension [20].

The latest research on BP monitors consists of fully eliminating the need for an inflatable cuff. One method that is being investigated is the use of photoplethysmography (PPG) for BP measurement [21,22]. Further investigation and validation are needed before implementation in clinical practice could be considered.

All this illustrates the evolution from office-based BP measurement to home-based BP measurement. Efficacy of home BP measurement was demonstrated in the TASMINH4 trial in which both self-monitoring BP at home and telemonitoring resulted in superior BP control compared to controls who performed in-office BP measurement only [23].

### 3.2. Physical Activity

Physical activity is an essential part of cardiovascular secondary prevention [9]. Moving physical activity for the cardiac patient to the digital era requires the following: monitoring parameters while performing exercise, monitoring amount of physical activity per day and offering active education, advice, and guidance about exercise [24,25].

Commercially available devices that track physical activity are abundant and include pedometers, activity bands and smartwatches. All recent smartphones have a built-in accelerometer which can be used to measure physical activity. Smartphone applications that register and track physical activity are widely used [26]. For the cardiac patient it is important that used devices are safe and validated. Many studies validate individual devices, and validation reference standards are proposed [27,28]. Algorithms to perform a validated 6-min walking test at home using a smartphone are being studied [29]. Delivery of cardiac rehabilitation to patients in the digital era includes offering home-based cardiac rehabilitation. An example of a smartphone-based rehabilitation program was given in the SMART-CR/SP trial [25]. This study demonstrated a significantly greater 6-min walking distance at 2 months and 6 months compared to controls.

CIEDs all contain rate response technology, which monitors physical activity to be able to increase the paced heart rate when the patient is performing exercise. Physical activity is measured by built-in sensors including accelerometers, pulmonary impedance sensors to measure minute ventilation, cardiac contractility sensors and blood temperature sensors [30]. Current research focuses on integrating this data with physical activity monitoring for cardiac rehabilitation purposes, and even for predicting decompensation of heart failure, rehospitalisation, and death [31,32].

For the cardiac patient, the digital future will involve better and more convenient monitoring of physical activity via ever smaller devices, as well as integrated telerehabilitation, for example via smartphone applications.

### 3.3. Weight Loss Management

Current guidelines on cardiovascular prevention recommend weight loss in patients with a BMI of >25 kg/$m^{2}$. Guidelines recommend achieving this by means of lifestyle changes (healthy diet and physical activity) and, if insufficient, pharmacologic therapy (orlistat, liraglutide). If the BMI is >40 kg/$m^{2}$ bariatric surgery can be recommended [9].

Digital technology through smartphone applications and wearables currently mainly focusses on physical activity and nutrition [33]. Nutritional education and recording are already widely used. The burden of manual input is currently considered the main disadvantage. Many apps focus on increasing usability by including features such as bar-code scanning, remembering recent food inputs, photo entry, auto-completion of text, and even recognizing food items from uploaded photos [34].

New developments include using artificial intelligence (AI), for example as a conversational tool serving as a digital weight loss coach as described by Stein et al. [35]. In this study, an AI-driven smartphone application can start conversations about diet and physical activity with the user. The researchers conclude that the use of an AI health coach is associated with weight loss comparable to in-person lifestyle interventions.

### 3.4. Diabetes Management

Glycaemic management is important for risk control in secondary cardiovascular prevention in people with diabetes mellitus [9]. Discussing all the recent evolutions in diabetes monitoring is beyond the scope of this viewpoint.

It can be summarized that while up to recently intermittent capillary finger-prick glucose monitoring with manual adjustment of insulin dose was common practice, there has already been a tremendous evolution in usage of continuous glucose monitoring through subcutaneous glucose sensors, smart insulin pens, automatic pumps and decision support systems [36]. Development is ongoing and might result in a, possibly AI-supported, “closed-loop” system in which all monitored data (e.g., glucose monitoring, physical activity, carbohydrates consumed, blood pressure, heart rate) is brought together and glucose monitoring and insulin therapy are fully automated without the need for interference by the patient or caregiver [36].

Development in HbA1c biosensors is also ongoing, and it is expected that in the future patients will have access to personal point-of-care HbA1c measurements that can be streamed to smartphones [37].

### 3.5. Smoking Cessation

Smoking cessation is the most cost-effective strategy for cardiovascular disease prevention. Current means of encouraging smoking cessation is through cognitive behavioural therapy as well as pharmacological therapy (nicotine replacement therapy, varenicline, bupropion) [9]. All of these methods rely heavily on health professionals initiating the treatment.

Smoking cessation solutions in the field of digital health currently mainly include smartphone applications. While many applications exist, only some are considered validated. Evidence-based behaviour change techniques include: supporting identity change, rewarding abstinence, changing routines and advising on medication use [38]. Recent studies combined the use of smartphone applications for behaviour change with exhaled carbon monoxide (CO)-sensors and dedicated programs with a human coach that follows up on the smartphone program [39,40]. Another type of sensor that is under development includes the “hand-to-mouth” wearable movement sensor for detecting smoking movements [41,42]. AI is used for predicting smoking cessation treatment outcomes [43] and is used in an AI-powered chatbot that aids in a behavioural change program [44].

Further research is needed to conclude about the relative effectiveness of all these techniques, but preliminary results are promising.

### 3.6. Medication Adherence

Medication adherence is an important part of secondary prevention and treatment of heart disease, as low medication adherence is associated with an increased risk for subsequent cardiovascular events [45]. Digital health is being explored to offer solutions to increase medication adherence, for example through smart pillboxes and digital pills. Smart pillboxes are pillboxes that are linked mostly to smartphone applications to monitor patient adherence. Digital pills are drug-device combinations that transmit a signal as soon as the pill comes into contact with gastric acids. While currently experimental, these techniques might be used in the future to monitor patient adherence [46].

## 4. Sensors in Heart Failure

Heart failure is a chronic condition, and one of the most challenging issues is to reduce hospital admission and readmission rates for worsening heart failure. Management during a hospital admission typically involves intensive parameter and blood sample monitoring and higher-than-usual dosage of medical treatment, for example with intravenous diuretics. Monitoring and treating heart failure could thus potentially benefit from a remote patient management approach. In recent years, extensive research in telemonitoring and remote treatment of heart failure has been conducted.

The most recent large trial that demonstrated the efficacy of telemonitoring in heart failure is the TIM-HF2 trial. Patients with New York Heart Association (NYHA) class II-III heart failure with a hospitalisation within the last 12 months were included in a structured remote patient management intervention which consisted of daily transmission of parameters (body weight, blood pressure, heart rate, heart rhythm, peripheral capillary oxygen saturation, and self-rated health status) using a telemonitoring system consisting of a three-channel electrocardiogram (ECG), a blood pressure measuring device and weighing scales connected to a mobile phone. The intervention reduced the percentage of days lost to unplanned cardiovascular hospital admission and all-cause mortality [6].

In recent years, invasive as well as non-invasive sensors have been developed to monitor patients with heart failure. Telemonitoring using CardioMEMS, a wireless implantable device that monitors pulmonary artery pressure, was shown to significantly lower hospital admission rates in heart failure NYHA class III [5]. In the LINK-HF study a non-invasive, disposable multisensor patch was used to measure ECG, skin impedance, temperature, and accelerometry to derive information about heart rate, heart rate variability, arrhythmia burden, respiratory rate, gross activity, walking, sleep, body tilt, and body posture. In this study it was shown that multisensor telemonitoring could provide accurate early detection of impending rehospitalisation in NYHA class II-IV [7].

Telemonitoring via CIEDs is also establishing its role in heart failure management. In the IN-TIME trial it was shown that multiparameter telemonitoring via implantable cardioverter-defibrillators (ICDs) and cardiac resynchronisation defibrillators (CRT-Ds) can significantly improve clinical outcomes for patients with heart failure [4]. Current evolutions in sensors as well as the mass rollout of 5G internet will likely enable hospitalisation and intensive care monitoring of patients with acute heart failure at home, both of which are currently being studied [47,48].

All this sensor-generated data is ideal for developing AI-algorithms. In the LINK-HF study, machine learning analytics was used to analyse multisensor data. Other recent developments of AI in heart failure include the interpretation of imaging data, predicting indication for and outcome after CRT implantation and using visually normal ECGs for electrolyte monitoring and predicting development of AF and heart failure [49].

## 5. Sensors in Arrhythmia Detection and Monitoring

An extensive review of digital technology in arrhythmia detection and treatment is out of the scope of this paper. A focused summary of detection and secondary prevention of AF and malignant arrhythmias will be provided here.

Ventricular malignant arrhythmias, including ventricular tachycardia (VT) and ventricular fibrillation (VF), are arrhythmias that, when untreated, can lead to sudden cardiac death [50]. The role of CIEDs in detecting, monitoring and, in the case of ICDs, treating ventricular arrhythmias is well-established [50]. Smartwatch-enabled diagnosis of ventricular arrhythmias using PPG is currently not possible, and reports of diagnosis of ventricular tachycardia (VT) using smartwatch ECGs are limited to case reports [51].

Concerning AF, three main methods for detection have been widely described, being PPG using smartphone cameras and smartwatches, single-lead ECG using a smartwatch or smartphone-compatible device and, less widespread, microelectromechanical sensors (MEMS) that detect cardiogenic movement when a smartphone is placed on the chest [52]. All of these techniques have shown good sensitivity and specificity, and both PPG-based devices (including FibriCheck) and ECG-based devices (including Apple Watch) have been approved for medical use by the United States Food and Drug Administration [52]. The Apple Heart study demonstrated feasibility of large-scale screening for AF [8]. While technology keeps evolving, the main debate has now shifted to answering the questions of which populations to screen, using what screening methodology, for how long to screen and what duration or burden of AF is sufficient to justify initiation of anticoagulation therapy [53,54].

## 6. The Internet of People and Digital Twins

While most current sensors require active engagement of the end-user, i.e., the patient, research is being performed on what is called the internet of people in which digital health gets more incorporated into everyday life and the need for active interaction with technology is reduced [55,56]. Biosensors are investigated that could be incorporated in unobtrusive places such as smart mirrors [57], smart clothing [58], smart vehicles [59], smart contact lenses [60], and even smart sanitation [61]. Voice recordings are increasingly investigated as biomarkers for coronary artery disease as well as heart failure decompensation [62,63,64]. If further developed, these promising technologies could offer an interesting track also in the field of telemonitoring and telerehabilitation.

Large-scale data collection together with improvement in routine diagnostics might lead to the ability of constructing of a ‘digital twin’ of a patient. The concept of a digital twin originates in engineering, and in healthcare it translates to a digital copy of the patient that simulates all structural and physiological characteristics. Heart function, exercise capacity, and medication tolerance could be simulated in very precise models of the patient, without the risk of harming the patient [65].

## 7. Barriers for Implementation

Digital health will be part of a promising future of medicine. Digital health however is not just about technology, and important barriers need to be addressed before the newest advances can be implemented in everyday medicine. In all facets of healthcare—patients, physicians, healthcare providers, legislators, and industry—certain barriers impede widespread implementation of technology [66].

Important factors that inhibit technology adoption in patients include lack of digital literacy and health literacy, lack of usability of technology and concerns about privacy and data confidentiality. Physicians and healthcare providers are often hesitant in adopting technology due to time constraints—often when digital health is not well incorporated in conventional workflow—lack of infrastructure and lack of digital training. Third, technological issues like interoperability are hampering further development of big data analysis and daily practical use of data. Lastly, there is a need for a clearly defined legal and ethical framework, including reimbursement policy, in which legal authorities have a clear role [49,66,67].

Research is increasingly being performed on technology interoperability. In a recent study a platform was developed for aggregating data from multiple sources, such as electronic health records (EHRs), pharmacies, personal digital devices and patient-reported outcome measures into one patient-centred platform for clinical as well as research purposes [68]. Another example is given by Aarestup et al., who put forward recommendations for using cloud technology through an open and decentralized infrastructure for data sharing and analysis for health research across the EU [69]; similar approaches could be thought of also for future patient-centred EHRs. On a regulatory level, The European Commission has recently adopted a Recommendation on a European Electronic Health Record exchange format that includes principles and technical specifications for the cross-border exchange of health data [70]. Examples of company-driven initiatives include the Philips Healthsuite Digital Platform that aims to integrate healthcare data from multiple sources in one patient-centred platform [71] and Yidu Cloud, a big-data platform for hospitals which has provided cloud services and data processing for more than 700 hospitals in China [72].

The Covid-19 pandemic has forced the healthcare system to review current practice, and digitisation of medicine has gained important momentum in all parties involved [73]. Patients were forced to use telecommunication tools for contacting family members and to work from home and thus had to become more digitally fluent [74], physicians utilised digital technology to monitor and contact patients, and government authorities had to review privacy policies to enable data analysis of important pandemic-related parameters [75].

In the coming years all parties should use this momentum to avoid taking steps backwards in the “post-Covid” era and to continue legal framework design and technological development. One of the main challenges will be to keep the patient in the centre of development to achieve a patient-centred digital healthcare system.

## 8. The Ideal Digital Tool for Cardiac Rehabilitation and Secondary Prevention

Cardiac rehabilitation is one of the modalities in medicine that could greatly benefit from digital health integration. As mentioned before, current secondary prevention compliance and cardiac rehabilitation participation rates are low, and optimisation is urgently required.

In the future, we envision a healthcare system that assembles a broad spectrum of patient data (including parameters, movement, ECGs, lab tests, genomics, etc.) into one electronic health record to be directly analysed by an AI-algorithm. Such a system could pinpoint people at high risk for cardiovascular disease and then puts maximal effort in preventing cardiac events from happening. In such a system, cardiac disease burden might be significantly lower. In those people that do suffer from cardiac pathology, rehabilitation will be a completely different world compared to now.

An ideal program should be capable of being delivered fully to the patients’ homes, either as add-on or as replacement of traditional centre-based cardiac rehabilitation. A multitude of sensors (invasive and non-invasive) can monitor parameters such as heart rate, blood pressure and weight, physical activity, pulmonary artery pressure, thoracic wall impedance, but also behaviour such as medication adherence and movements associated with active smoking. Smart tools can make sure that data input is automatic to maximize patient comfort, and interoperability of technological systems ensures user-friendliness also for the healthcare provider. AI-driven algorithms can offer decision support on medication and physical activity prescription and could be enabled to run autonomously in most instances. In this way, highly personalised medicine is offered to the patient while achieving a low burden on the healthcare system. Secondary prevention could be run by an almost fully automated, individualised AI-driven virtual health coach that we currently envision in the form of a smartphone application but could be integrated in any future wearable hardware.

All this should be integrated in a system that is backed-up by a human team of cardiac rehabilitation healthcare providers. While routine matters run fully automatic, a specialised team can offer targeted care where a human touch is needed. Algorithms can alert when something seems not right, urging healthcare workers, for example a specialised nurse, a dietician or a psychologist, to contact those patients in whom help seems most needed. Video technology can enable swift contact also for those living far away from healthcare facilities. If wanted, patients could be able to make contact with peers to exchange experiences, participate in group sessions and support each other.

Most importantly, the ideal sensor puts patients first. While registering data, the patient should not have to notice the sensor, should not experience any discomfort, and should not have to manually input data.

Privacy and data protection are also of key importance in this sense, and an ideal healthcare system should evolve to use sensors that register data that is the property of the patient and that is collected in a patient-based system. Ideally, a framework for a patient-based platform should be defined, in which high interoperability and data safety standards are defined by governments, and in which all data should be translated to the same interoperable language. Companies can then still each design their own interface. As such, patients, caregivers, hospitals, and companies can all speak the same digital language while using hardware and software to their own preference. Data should be owned by the patient. The patient can share the data to the EHR used by the caregiver in a medical contact setting. From the EHR the data could be subjected to AI analysis. Only if the patient consents, the data can be shared with research institutions or with healthcare companies.

## 9. Conclusions

If we continue development and integration of technology with the patient in mind and in dialog with patients and caregivers, the digital evolution could evolve in an optimal way. Cardiac care and cardiac rehabilitation will be able to take leaps forward. Aided by digital technology, we will be able to offer high-quality, affordable, personalised healthcare in a patient-centred way.

## Data Availability

No new data were created or analyzed in this study. Data sharing is not applicable to this article.

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
