# Peer review of "Digital Health in Cardiac Rehabilitation and Secondary Prevention: A Search for the Ideal Tool"

_sensors, 2020, doi:10.3390/s21010012_

Round 1

Reviewer 1 Report

I appreciated the author for giving me the chance to read this article. As the authors stated in the manuscript, one of the biggest problem for the implementation of such sensors may be that we need a good platform which can organize and coordinate multiple sensors and manage patients condition. If the authors have any idea how to develop such a platform, please describe it in the manuscript. 

Author Response

The authors thank the reviewer for this comment and agree that it is indeed desirable to elaborate on this interesting problem.

  • In the section “Barriers for implementation” the following text was added:
    • “Research is increasingly being performed on technology interoperability. In a recent study a platform was developed for aggregating data from multiple sources, such as electronic health records (EHRs), pharmacies, personal digital devices and patient-reported outcome measures into one patient-centred platform for clinical as well as research purposes (69). Another example is given by Aarestup et al., who put forward recommendations for using cloud technology through an open and decentralized infrastructure for data sharing and analysis for health research across the EU (70); similar approaches could be thought of also for future patient-centred EHRs. On a regulatory level, The European Commission has recently adopted a Recommendation on a European Electronic Health Record exchange format that includes principles and technical specifications for the cross-border exchange of health data (71). Examples of company-driven initiatives include the Philips Healthsuite Digital Platform that aims to integrate healthcare data from multiple sources in one patient-centred platform (72) and Yidu Cloud, a big-data platform for hospitals which has provided cloud services and data processing for more than 700 hospitals in China (73).”
  • In the paragraph “The ideal digital tool for cardiac rehabilitation and secondary prevention” the end of the text was adjusted to:
    • “Ideally, a framework for a patient-based platform should be defined, in which high interoperability and data safety standards are defined by governments, and in which all data should be translated to the same interoperable language. Companies can then still each design their own interface. As such, patients, caregivers, hospitals and companies can all speak the same digital language while using hardware and software to their own preference. Data should be owned by the patient. The patient can share the data to the EHR used by the caregiver in a medical contact setting. From the EHR the data could be subjected to AI analysis. Only if the patient consents, the data can be shared with research institutions or with healthcare companies.”

Reviewer 2 Report

The article is a review of the digital tools and techniques that are currently applied or tested in cardiac rehabilitation and secondary prevention. In the first part, sensors that can be used for telemonitoring in hypertension, physical activity, weight loss management, diabetes management, smoking cessation and medication adherence are covered. Moreover, sensors in heart failure and the barriers of implementation are discussed. In the last part of the article, "The ideal digital tool for cardiac rehabilitation and secondary prevention", authors envision a healthcare system that collects and analyses multiple data of one huge electronic health record.

However, the selection of the topics detailed in the review is arbitary, many significant cardiac diseases are mentioned, while others that can also be important  targets of telemonitorization - like atrial fibrillation or malignant ventricular arrhythmias - are not. Moreover, the methods of data collection for the review, the searched databases etc. are not clear.

In conclusion, the article is an interesting review of the huge potential that lies in e-health care systems and also contains important issues about the possibilities of future developments in this area. I suggest the authors to also add a methods section and to supplement the subsections of the article especially with some important other possibilities of telemonitoring.

Author Response

The authors thank the reviewer for the clear and interesting comments.

  • The Introduction was adapted to include the aim of the paper, in which it is explained that because the focus is on cardiac rehabilitation, the focus is also on the most common pathologies in cardiac rehabilitation, being ischemic heart disease and heart failure:
    • “The aim of this narrative review is to offer a perspective on the current use of digital technology in cardiac rehabilitation and secondary prevention. As most benefit of cardiac rehabilitation has been shown in patients with ischemic heart disease and heart failure (15,16), the focus of this paper will be on these patient populations. The viewpoint also offers a vision of what ideal secondary prevention and rehabilitation aided by digital tools might look like in the future.”
  • A methods section was added in which the authors now specify that this is indeed a narrative review:
    • “2. Methods:
    • This viewpoint is written as a narrative review. The databases PubMed, Embase and Web of Science were used. Keywords were defined corresponding to the subjects discussed and included cardiac rehabilitation, heart failure, digital health, sensors, among other keywords for the specific subtopics.”
  • The authors also agree with the reviewer that indeed arrhythmia detection and monitoring is an important part in telemonitoring and secondary prevention, and a section on this topic was thus added:
    • “5. Sensors in arrhythmia detection and monitoring.
    • An extensive review of digital technology in arrhythmia detection and treatment is out of the scope of this paper. A focused summary of detection and secondary prevention of AF and malignant arrhythmias will be provided here.
    • Ventricular malignant arrhythmias, including ventricular tachycardia (VT) and ventricular fibrillation (VF), are arrhythmias that, when untreated, can lead to sudden cardiac death (51). The role of CIEDs in detecting, monitoring and, in the case of ICDs, treating ventricular arrhythmias is well-established (51). Smartwatch-enabled diagnosis of ventricular arrhythmias using PPG is currently not possible, and reports of diagnosis of ventricular tachycardia (VT) using smartwatch ECGs are limited to case reports (52).
    • Concerning AF, three main methods for detection have been widely described, being PPG using smartphone cameras and smartwatches, single-lead ECG using a smartwatch or smartphone-compatible device and, less widespread, microelectromechanical sensors (MEMS) that detect cardiogenic movement when a smartphone is placed on the chest (53). All of these techniques have shown good sensitivity and specificity, and both PPG-based devices (including FibriCheck) and ECG-based devices (including Apple Watch) have been approved for medical use by the United States Food and Drug Administration (53). The Apple Heart study demonstrated feasibility of large-scale screening for AF (8). While technology keeps evolving, the main debate has now shifted to answering the questions of which populations to screen, using what screening methodology, for how long to screen and what duration or burden of AF is sufficient to justify initiation of anticoagulation therapy (54,55).”

Reviewer 3 Report

The manuscript titled “Digital health in cardiac rehabilitation and secondary prevention: search for the ideal tool” deals with the perspective on current use of digital health technologies in cardiac rehabilitation, heart failure and secondary prevention. The authors introduced the study of the state of the art and offer the good viewpoint. However, I think the contents of the manuscript are common and needs more novelty. In order for the manuscript to be published in the Sensors (to maintain the high quality of the Sensors), it seems to the additional elements are necessary to make readers highly interested.

Author Response

The authors thank the reviewer for the comment and understand the concern about novelty in the previous version of the manuscript. The authors have now revised the article to include new specific examples on what is considered cutting-edge technology not yet included in thorough clinical research. This overview is given in a new paragraph:

"6. The internet of people and digital twins

While most current sensors require active engagement of the end-user, i.e. the patient, research is being performed on what is called the internet of people in which digital health gets more incorporated into everyday life and the need for active interaction with technology is reduced (56,57). Biosensors are investigated that could be incorporated in unobtrusive places such as smart mirrors (58), smart clothing (59), smart vehicles (60), smart contact lenses (61) and even smart sanitation (62). Voice recordings are increasingly investigated as biomarkers for coronary artery disease as well as heart failure decompensation (63–65). If further developed, these promising technologies could offer an interesting track also in the field of telemonitoring and telerehabilitation.

Large-scale data collection together with improvement in routine diagnostics might lead to the ability of constructing of a ‘digital twin’ of a patient. The concept of a digital twin originates in engineering, and in healthcare it translates to a digital copy of the patient that simulates all structural and physiological characteristics. Heart function, exercise capacity and medication tolerance could be simulated in very precise models of the patient, without the risk of harming the patient (66)." 

Also, in section 7 (Barriers for implementation) and 8 (The ideal digital tool for cardiac rehabilitation and secondary prevention) we now elaborate on current research about future electronic health records, interoperability and multiple source data integration.

"7. Barriers for implementation

[…] Research is increasingly being performed on technology interoperability. In a recent study a platform was developed for aggregating data from multiple sources, such as electronic health records (EHRs), pharmacies, personal digital devices and patient-reported outcome measures into one patient-centred platform for clinical as well as research purposes (69). Another example is given by Aarestup et al., who put forward recommendations for using cloud technology through an open and decentralized infrastructure for data sharing and analysis for health research across the EU (70); similar approaches could be thought of also for future patient-centred EHRs. On a regulatory level, The European Commission has recently adopted a Recommendation on a European Electronic Health Record exchange format that includes principles and technical specifications for the cross-border exchange of health data (71). Examples of company-driven initiatives include the Philips Healthsuite Digital Platform that aims to integrate healthcare data from multiple sources in one patient-centred platform (72) and Yidu Cloud, a big-data platform for hospitals which has provided cloud services and data processing for more than 700 hospitals in China (73). […]

8. The ideal digital tool for cardiac rehabilitation and secondary prevention

[…] Ideally, a framework for a patient-based platform should be defined, in which high interoperability and data safety standards are defined by governments, and in which all data should be translated to the same interoperable language. Companies can then still each design their own interface. As such, patients, caregivers, hospitals and companies can all speak the same digital language while using hardware and software to their own preference. Data should be owned by the patient. The patient can share the data to the EHR used by the caregiver in a medical contact setting. From the EHR the data could be subjected to AI analysis. Only if the patient consents, the data can be shared with research institutions or with healthcare companies."

Reviewer 4 Report

Dear Authors,

I read your manuscript with great interest and I feel that it holds merit for a potential broader audience.

However, I came upon a few points that must be addressed:

-) Title: change to "[...] A search for the [...]"

-) Abstract: The first sentence reads confusing, please clarify.

-) Abstract: The second sentence is very general and does not hold any important information, try to rephrase.

-) Abstract: The last sentence reads quite poetic and not scientific enough, please rephrase.

-) Overall manuscript: You incorporated some very poetic-sounding phrases and sentences that do not meet the scientific language criteria expected by an international audience. Try to go through the whole manuscript and rephrase respective passages (for example, the first two sentences of the Introduction).

-) Introduction, lines 25-28: please cite.

-) Introduction, lines 40-43: please cite.

-) Introduction, lines 48-50: also consider other forms of risk prevention, for example mention and cite https://pubmed.ncbi.nlm.nih.gov/31933684/

-) Lines 57-60: please cite.

-) Line 67: I would change the sentence to "[...] research ON BP monitors [...]"

-) Lines 76-79: please cite.

-) Lines 117-118: please cite.

-) Lines 120-123: please cite.

-) Lines 157-162: please cite.

-) Line 165: which structured remote patient management? Please describe briefly.

-) Liness 194-195: please cite (also, if the whole paragraph is ref. 40-42)

-) I would add a subheading named "Conclusion" at the end, as most readers will expect to find one.

Author Response

The authors thank the reviewer for his/her careful reading of the manuscript and his/her constructive remarks. Please find below the point-by-point response to all comments.

Dear Authors,

I read your manuscript with great interest and I feel that it holds merit for a potential broader audience.

However, I came upon a few points that must be addressed:

-) Title: change to "[...] A search for the [...]"

The title was adjusted according to this comment.

-) Abstract: The first sentence reads confusing, please clarify.

This sentence was changed to: “Digital health is becoming more integrated in daily medical practice.” 

-) Abstract: The second sentence is very general and does not hold any important information, try to rephrase.

This sentence was changed to:

    • “In cardiology, patient care is already moving from the hospital to the patients’ homes, with large trials showing positive results in the field of telemonitoring via CIEDs, monitoring of pulmonary artery pressure via implantable devices, telemonitoring via home-based non-invasive sensors and screening for atrial fibrillation via smartphone and smartwatch technology.”

-) Abstract: The last sentence reads quite poetic and not scientific enough, please rephrase.

The authors adapted this by leaving out the non-contributing words “very human”. It now reads:

    • “Aided by digital technology, a future could be realised in which we are able to offer high-quality, affordable, personalised healthcare in a patient-centred way.”

-) Overall manuscript: You incorporated some very poetic-sounding phrases and sentences that do not meet the scientific language criteria expected by an international audience. Try to go through the whole manuscript and rephrase respective passages (for example, the first two sentences of the Introduction).

The authors thank the reviewer for this comment. The manuscript was revised and the following sentences were changed and now read:

    • Introduction, sentence 1 and 2: “In recent years, digital health is becoming more incorporated in daily live in general, as well as in clinical practice in medicine.”
    • Barriers, line 217: "One of the main challenges will be to keep the patient in the centre of development to achieve a patient-centred digital healthcare system."
    • Closing paragraph: “If we continue development and integration of technology with the patient in mind and in dialog with patients and caregivers, the digital evolution could evolve in an optimal way. Cardiac care and cardiac rehabilitation will be able to take leaps forward. Aided by digital technology, we’ll be able to offer high-quality, affordable, personalised healthcare in a patient-centred way.” 

-) Introduction, lines 25-28: please cite.

The following citations were added:

    • Cowie MR, Bax J, Bruining N, Cleland JGF, Koehler F, Malik M, et al. E-Health: A position statement of the European Society of Cardiology. Eur Heart J. 2016;37(1):63–6.
    • Chakravartti J, Rao S V. Robotic Assisted Percutaneous Coronary Intervention: Hype or Hope? 1977;1–4.
    • Norgeot B. A call for deep-learning healthcare. Nat Med. 2019;25(January):14–5.

-) Introduction, lines 40-43: please cite.

The following citations were added:

    • Ruano-Ravina A, Pena-Gil C, Abu-Assi E, Raposeiras S, van ’t Hof A, Meindersma E, et al. Participation and adherence to cardiac rehabilitation programs. A systematic review. Int J Cardiol. 2016 Nov 15;223:436–43. Available from: https://doi.org/10.1016/j.ijcard.2016.08.120
    • Frederix I, Solmi F, Piepoli MF, Dendale P. Cardiac telerehabilitation: A novel cost-efficient care delivery strategy that can induce long-term health benefits. Vol. 24, European Journal of Preventive Cardiology. 2017. p. 1708–17.

-) Introduction, lines 48-50: also consider other forms of risk prevention, for example mention and cite https://pubmed.ncbi.nlm.nih.gov/31933684/

The following sentence and reference were added:

    • “In psychosocial management techniques such as breathing exercises and medication are often included (15).”
    • Schnaubelt S, Hammer A, Koller L, Niederdoeckl J, Kazem N, Spiel A, et al. Expert Opinion Meditation and Cardiovascular Health : What is the Link ? 2019;161–4.

-) Lines 57-60: please cite.

 The following citation was added:

    • Williams, B. et al. (2018) 2018 practice guidelines for the management of arterial hypertension of the European society of cardiology and the European society of hypertension ESC/ESH task force for the management of arterial hypertension, Journal of Hypertension. doi: 10.1097/HJH.0000000000001961.

-) Line 67: I would change the sentence to "[...] research ON BP monitors [...]"

The authors thank the reviewer for the comment, this was adapted to “The latest research on BP monitors[…]” 

-) Lines 76-79: please cite.

The following citations were added:

    • Piepoli, M. F. et al. (2016) ‘2016 European Guidelines on cardiovascular disease prevention in clinical practice’, European Heart Journal, 37(29), pp. 2315–2381. doi: 10.1093/eurheartj/ehw106.
    • Dorje, T. et al. (2019) ‘Smartphone and social media-based cardiac rehabilitation and secondary prevention in China (SMART-CR/SP): a parallel-group, single-blind, randomised controlled trial’, The Lancet Digital Health. The Author(s). Published by Elsevier Ltd. This is an Open Access article under the CC BY-NC-ND 4.0 license, 1(7), pp. e363–e374. doi: 10.1016/S2589-7500(19)30151-7.
    • Wang, W. and Jiang, Y. (2019) ‘The evolving mHealth-based cardiac rehabilitation’, The Lancet Digital Health. The Author(s). Published by Elsevier Ltd. This is an Open Access article under the CC BY-NC-ND 4.0 license, 1(7), pp. e326–e327. doi: 10.1016/S2589-7500(19)30155-4.

-) Lines 117-118: please cite.

The following citation was added:

    • Piepoli, M. F. et al. (2016) ‘2016 European Guidelines on cardiovascular disease prevention in clinical practice’, European Heart Journal, 37(29), pp. 2315–2381. doi: 10.1093/eurheartj/ehw106.

-) Lines 120-123: please cite.

The following citation was added:

    • Vettoretti M, Cappon G, Facchinetti A, Sparacino G. Advanced diabetes management using artificial intelligence and continuous glucose monitoring sensors. Sensors (Switzerland). 2020;20(14):1–18.

-) Lines 157-162: please cite.

The following citation was added:

    • Ponikowski, P. et al. (2016) ‘2016 ESC Guidelines for the diagnosis and treatment of acute and chronic heart failure’, European Heart Journal, 37(27), pp. 2129-2200m. doi: 10.1093/eurheartj/ehw128.

For the last sentences - “Monitoring and treating heart failure could thus potentially benefit from a remote patient management approach. In recent years, extensive research in telemonitoring and remote treatment of heart failure has been conducted.” – the examples are given in the following sentences with appropriate references.

-) Line 165: which structured remote patient management? Please describe briefly.

A description was added to the following sentence:

    • “The most recent large trial that demonstrated the efficacy of telemonitoring in heart failure is the TIM-HF2 trial. Patients with New York Heart Association (NYHA) class II-III heart failure with a hospitalisation within the last 12 months were included in a structured remote patient management intervention which consisted of daily transmission of parameters (body weight, blood pressure, heart rate, heart rhythm, peripheral capillary oxygen saturation and self-rated health status) using a telemonitoring system consisting of a three-channel electrocardiogram (ECG), a blood pressure measuring device and weighing scales connected to a mobile phone. The intervention reduced the percentage of days lost to unplanned cardiovascular hospital admission and all-cause mortality.” 

-) Liness 194-195: please cite (also, if the whole paragraph is ref. 40-42)

The following citation was added:

    • Frederix, I. et al. (2019) ‘ESC e-Cardiology Working Group Position Paper: Overcoming challenges in digital health implementation in cardiovascular medicine’, European Journal of Preventive Cardiology, 26(11), pp. 1166–1177. doi: 10.1177/2047487319832394. 

-) I would add a subheading named "Conclusion" at the end, as most readers will expect to find one.

The subheading was added before the last paragraph.

Round 2

Reviewer 3 Report

The authors have addressed my primary concern by adding the new specific examples (Sensors in arrhythmia detection and monitoring, The internet of people and digital twins, etc), and I feel the paper is now acceptable for publication.
